# Stockpiled personal protective equipment and knowledge of pandemic plans as predictors of perceived pandemic preparedness among German general practitioners

**Arno Stöcker** \*, **Ibrahim Demirer**, **Sophie Gunkel, Jan Hoffmann**, **Laura Mause, Tim Ohnhäuser, Nadine Scholten**

Faculty of Human Sciences, Institute of Medical Sociology, Health Services Research, and Rehabilitation Science, Faculty of Medicine and University Hospital, University of Cologne, Cologne, Germany

\* arno.stoecker@uk-koeln.de

## Abstract

### Background

The COVID-19 pandemic significantly changed the work of general practitioners (GPs). At the onset of the pandemic in March 2020, German outpatient practices had to adapt quickly. Pandemic preparedness (PP) of GPs may play a vital role in their management of a pandemic.

### Objectives

The study aimed to examine the association in the stock of seven personal protective equipment (PPE) items and knowledge of pandemic plans on perceived PP among GPs.

### Methods

Three multivariable linear regression models were developed based on an online cross-sectional survey for the period March–April 2020 (the onset of the pandemic in Germany). Data were collected using self-developed items on self-assessed PP and knowledge of a pandemic plan and its utility. The stock of seven PPE items was queried. For PPE items, three different PPE scores were compared. Control variables for all models were gender and age.

### Results

In total, 508 GPs were included in the study; 65.16% believed that they were very poorly or poorly prepared. Furthermore, 13.83% of GPs were aware of a pandemic plan; 40% rated those plans as beneficial. The stock of FFP-2/3 masks, protective suits, face shields, safety glasses, and medical face masks were mostly considered completely insufficient or insufficient, whereas disposable gloves and disinfectants were considered sufficient or completely sufficient. The stock of PPE was significantly positively associated with PP and had the

**Data Availability Statement:** The authors have uploaded a minimal data set to Data Archiving and

Networked Services (DANS) which is available at
https://doi.org/10.17026/dans-z7a-b6p9.

**Funding:** This work is associated with the research
project "COVID GAMS" (https://www.covid-gams.
de) funded by the German Federal Ministry of
Education and Research (BMBF 01KI2099, https://
www.bmbf.de/bmbf/en/). NS received a grant from
which AS is partly paid. The funding source had no
role in the design, conduct, or reporting of this
study; or in the decision to submit the manuscript
for publication.

**Competing interests:** The authors have declared
that no competing interests exist.

largest effect on PP; the association of the knowledge of a pandemic plan was significant
but small. PPE scores did not vary considerably in their explanatory power. The assessment
of a pandemic plan as beneficial did not significantly affect PP.

## Conclusion

The stock of PPE seems to be the determining factor for PP among German GPs; for
COVID-19, sufficient masks are the determining factor. Knowledge of a pandemic plans
play a secondary role in PP.

## Introduction

The coronavirus disease 2019 (COVID-19) pandemic took Germany and many other coun-
tries by surprise in 2020. Within a few weeks, social and work life changed. The healthcare sec-
tor was one of the most affected areas [1–3]. Pandemic plans were activated [4], and
emergency measures were taken in hospitals and intensive care units to treat a large number of
patients with COVID-19 [5]. However, general practitioners (GPs) are often the first to have
contact with potential patients with COVID-19 [3,6] and the majority of patients with
COVID-19 –mostly with mild and moderate symptoms [7,8]–are treated in GP practices.
Apart from the additional workload of treating patients with COVID-19, GPs have to maintain
regular primary health care [2,3,9,10]. General practitioners have been facing multiple chal-
lenges during the COVID-19 pandemic, such as a high risk of being infected by severe acute
respiratory syndrome coronavirus 2 (SARS-CoV-2) themselves [11], including the risk of mor-
tality [12] and psychological impacts with regard to work and personal life [13]. Overall, pan-
demic preparedness (PP) is an important factor in being better able to manage the challenges
of a pandemic [14,15].

At the beginning of the COVID-19 pandemic, many personal protective equipment (PPE)
items were in short supply in Germany and worldwide [16,17], both for private use [18] and in
the healthcare sector [19,20]. Specifically, medical masks and FFP-2/3 masks were in high
demand and short supply [20]. As Germany has a federated and self-governing system in the
healthcare sector with 17 self-regulatory regional organization for the outpatient sector, clear
division of responsibilities was often absent [21]. Many physicians complained about a general,
long-lasting shortage of PPE [22,23].

Prior to the outbreak of the COVID-19 pandemic, a variety of pandemic plans existed in
Germany [24]. For example, the Robert Koch Institute, a German federal government agency
responsible for disease control and prevention, established a national influenza pandemic plan
(last update 2016/2017) [25]. On March 4, 2020, a supplement to the national pandemic plan
regarding COVID-19 was published [26]. On March 13, 2020, the national pandemic plan has
been activated [27]. In addition, the Kassenärztliche Bundesvereingung (German Association
of Statutory Health Insurance Physicians) had published a document regarding influenza pan-
demic "Risk Management in Medical Practices" in 2008, which is specifically aimed at the out-
patient sector [28]. Furthermore, each of the 16 German states has its own pandemic plan, and
several cities have also established their own pandemic plans [24,29]. In the event of an influ-
enza pandemic, German pandemic plans ensure priority of outpatient treatment [24].

While the necessity of sufficient knowledge [30] and adequate PPE in general [7–9] have
been recognized for protecting health care professionals and maintaining the operation of
medical facilities during a pandemic, different PPE items have been considered and compared

less often. Therefore, in this study, we aim to further investigate how stocked PPE items effect perceived PP. In addition to adequate PPE, pandemic plans are considered a firm cornerstone with regard to PP in the healthcare sector [31,32]. However, the effect of pandemic plans on GPs' personal PP has not been studied to our knowledge; other studies are limited to polling knowledge of pandemic plans and their utility [8]. Hence, in this study we investigated how the stock of seven PPE items and knowledge of a pandemic plan are associated with PP among German GPs.

## Method

### Design

This analysis is part of the research project "The COVID-19 crisis and its impact on the German ambulatory sector–the physicians' view" (COVID-GAMS). The study is based on an anonymous, online cross-sectional survey that is conducted at three different points in time in 2020 and 2021. The first survey was conducted in June–September 2020 retrospectively for the period March–April 2020, which corresponded with the peak of the first COVID-19 wave in Germany. During questionnaire conception and development, preliminary interviews were conducted with different representatives of the listed group of specialists. The questionnaire was subsequently tested by several physicians from different specialty groups who were not involved in the design. The questions relevant to this study can be accessed in German and English in the appendix (S1 Table).

### Participants and recruitment

A total of 18,000 outpatient physicians were invited to participate in the online survey: GPs (6,500), dentists (4,000), gynecologists (2,000), pediatricians (2,000), otolaryngologists (2,000), cardiologists (1,000), and gastroenterologists (500). The study population was selected to capture outpatient care during the Corona pandemic from the perspective of different medical disciplines. The address data for the random study sample was selected in collaboration with the National Association of Statutory Health Insurance Physicians. Invitations were sent by fax and e-mail, followed by three reminders at 2-week intervals. In addition, physicians were invited to participate in the survey via the project homepage (www.covid-gams.de) and various specialist associations. In this analysis, we examine only responses of GPs. The survey (including invitation letter, study and privacy information and questionnaire) was approved by the Ethics Committee of the University of Cologne (20–1169_1). The online survey was conducted anonymously, without directly collecting personal identifying information, so that only implicit consent had to be obtained in accordance with the ethics vote of the Ethics Committee of the University Hospital of Cologne. The terms and conditions of the study had to be agreed to in order to participate in the study. Participation in the survey could be terminated at any time. The possibility to pause the survey and continue it later was technically possible. Participation was voluntary for all participants. No expense allowance or payment was paid for participation.

### Measures

The focus of the analysis, examined as the dependent variable, was the following research question: "How prepared did you feel at your practice for a pandemic in early March?" Answers were given on a 5-point Likert scale (1 = very bad, 2 = bad, 3 = moderate, 4 = good, 5 = very good) and processed as numerical outcomes [33]. The first predictor was stockpiled PPE (FFP-2/3 masks, medical face masks, disposable gloves, hand and surface disinfectants, safety

glasses, protective suits, and face shields). The following answers could be given for the question "As of early March, what was your inventory of the following protective and hygienic materials?": not relevant, completely inadequate (1), inadequate (2), adequate (3), completely adequate (4). Responses with "not relevant" were excluded for further analysis. The second predictor of interest was knowledge of an epidemic or pandemic plan (yes/no). General practitioners who reported having knowledge of a pandemic plan were further asked whether those plans helped them manage during the COVID-19 pandemic (yes/no). On the basis of these responses, two binary-coded variables were developed (no = 0, yes = 1). Age (in 10-year increments) and gender of the participating GPs were included as control variables in each regression model. Gender has been found to affect PP in some studies in the past [34,35]. Early in the COVID-19 pandemic, there was strong evidence that advanced age has an effect on disease progression [36] and, hence, older GPs may have felt generally less prepared for the pandemic. Therefore, it was reasonable to include both variables in the model to control for any confounding effects. Because the study population of GPs was otherwise rather homogeneous, we refrained from including additional control variables.

### Statistical analysis

Three groups of multivariate regression models were used to examine the factors affecting perceived PP. In the first set of models, three different PPE scores were compared. We assumed that there is an underlying interplay between relevant PPE items, hence, different PPE scores were computed. If an essential PPE item is missing, the protection chain may be interrupted, so that even items that are actually sufficiently on hand cannot develop their full protective effect. Therefore, it seemed appropriate to combine PPE items into PPE scores. Three types of PPE scores were obtained: 1. a general PPE score with all PPE items combined, 2. an exploratively investigated optimized PPE score with those PPE items that showed a significant association with PP, and 3. a masks-only score with the two PPE mask types (FFP-2/-3 and medical masks) as masks provide the greatest protection against SARS-CoV-2 infection. For each type of score, the numeric responses given to each included PPE item were summed and divided by the number of PPE items. Thus, each PPE score ranges between 1 and 4. The optimized PPE score is an exploratory compiled value used to compare the goodness of the models with the general PPE and the masks score. In the second regression model, the association of knowledge of a pandemic plan on PP was examined. In addition, a potential interaction effect between knowledge of a pandemic plan and stocked PPE on perceived PP was explored. Knowledge of a pandemic plan may have an interaction effect with stocked PPE, as more PPE maybe stored if a pandemic plan is known or higher PP can be reported with the same level of stockpiled PPE if a pandemic plan is known. The third model investigated whether a pandemic plan rated as beneficial was associated with perceived PP. Again, a potential interaction effect between a pandemic plan viewed as beneficial and PPE in storage was included in the model to test interactions similar to those described in model 2. Data preparation (tidyverse package [1.3.0]) and analysis (lessR package [3.9.9] and psych package [2.0.12]) were performed in R (version 4.03) and R Studio (version 1.3.1093).

## Results

### Sample

In total, 1,703 physicians participated in the first survey, including 535 GPs. Of these, 508 GPs responded to the relevant item on PP. In total, 265 male, 242 female, and one non-binary GP participated (Table 1); of these, 11.05% were 31–40 years old, 25.44% were 41–50 years old, 40.83% were 51–60 years old, and 22.68% were older than 60 years.

**Table 1. General practitioners' (n = 508) characteristics and pandemic preparedness, personal protective equipment, and knowledge on a pandemic plan and its utility.**

| Variables | n (%) |
|---|---|
| Pandemic preparedness (n = 508) | |
| *very poor* | 134 (26.38) |
| *poor* | 197 (38.78) |
| *partly* | 132 (26.98) |
| *good* | 34 (6.67) |
| *very good* | 11 (2.17) |
| *missings* | - |
| FFP-2/3 Masks (n = 507) | |
| *not relevant* | 4 (0.79) |
| *completely insufficient* | 315 (62.13) |
| *insufficient* | 125 (24.65) |
| *sufficient* | 52 (10.26) |
| *completely sufficient* | 11 (2.17) |
| *missings* | 1 |
| Mouth and nose protection (n = 508) | |
| *not relevant* | 1 (0.20) |
| *completely insufficient* | 118 (23.23) |
| *insufficient* | 214 (42.13) |
| *sufficient* | 141 (27.76) |
| *completely sufficient* | 34 (6.69) |
| *missings* | - |
| Disposable gloves (n = 508) | |
| *not relevant* | - |
| *completely insufficient* | 12 (2.36) |
| *insufficient* | 61 (12.01) |
| *sufficient* | 305 (60.04) |
| *completely sufficient* | 130 (25.59) |
| *missings* | - |
| Hand and surface disinfectants (n = 508) | |
| *not relevant* | - |
| *completely insufficient* | 22 (4.33) |
| *insufficient* | 105 (20.67) |
| *sufficient* | 276 (54.33) |
| *completely sufficient* | 105 (20.67) |
| *missings* | - |
| Safety glasses (n = 508) | |
| *not relevant* | 9 (1.77) |
| *completely insufficient* | 253 (49.80) |
| *insufficient* | 128 (25.20) |
| *sufficient* | 95 (18.70) |
| *completely sufficient* | 23 (4.53) |
| *missings* | |
| Protective suits (n = 508) | |
| *not relevant* | 9 (1.77) |
| *completely insufficient* | 305 (60.04) |
| *insufficient* | 130 (25.59) |
| *sufficient* | 52 (10.24) |
| *completely sufficient* | 12 (2.36) |

(*Continued*)

**Table 1.** (Continued)

| Variables | n (%) |
|---|---|
| *missings* | |
| Face shields (n = 507) | |
| *not relevant* | 57 (11.24) |
| *completely insufficient* | 331 (65.29) |
| *insufficient* | 81 (15.98) |
| *sufficient* | 30 (5.92) |
| *completely sufficient* | 8 (1.58) |
| *missings* | 1 |
| Prior knowledge of any pandemic plan (n = 506) | |
| *no* | 436 (86.17) |
| *yes* | 70 (13.83) |
| *missings* | 2 |
| if yes: | |
| Helpfulness of pandemic plan (n = 70) | |
| *no* | 42 (60.00) |
| *yes* | 28 (40.00) |
| *missings* | - |
| Age (n = 507) | |
| *30 years and younger* | - |
| *31 to 40 years* | 56 (11.05) |
| *41 to 50 years* | 129 (25.44) |
| *51 to 60 years* | 207 (40.83) |
| *older than 60 years* | 115 (22.68) |
| *missings* | 1 |
| Gender (n = 508) | |
| *male* | 265 (52.16) |
| *female* | 242 (47.64) |
| *none-binary* | 1 (0.00) |
| *missings* | - |

Nearly two-thirds of GPs believed that they and their practice were poorly or very poorly prepared for a pandemic; only 8.84% reported that they were well or very well prepared. In terms of PPE stock, GPs reported that FFP-2/3 masks (89.78%), protective suits (85.63%), face shields (81.27%), safety glasses (75.00%), and medical face masks (65.36%), respectively, were completely insufficient or insufficient, whereas GPs reported that disposable gloves (85.63%) and hand and surface disinfectants (75.00%) were sufficient or completely sufficient at the beginning of March 2020. Cronbach's alpha for the seven PPE items was .81 (CI: .78; .83). There was no collinearity between different PPE items as no individual variance inflation factors did exceed 2 (Table 2). A high number of GPs (86.17%) had no knowledge of a pandemic plan; of the 70 GPs who had knowledge of such a plan, a slight majority (60%) rated such plans as not beneficial for the SARS-CoV-2 pandemic.

## Multivariable linear regression models

Individuals with no information on age or gender or categories with less than three individuals per group were excluded from further investigation for statistical reasons (n = 2). In total,

**Table 2. Collinearity: Variance inflation factor and tolerance for personal protective equipment items.**

|  | VIF | Tolerance |
|---|---|---|
| FFP-2/3 masks | 1.750 | .571 |
| medical masks | 1.500 | .667 |
| surgical gloves | 1.917 | .522 |
| hand and surface disinfectants | 1.939 | .516 |
| safety glasses | 1.739 | .575 |
| protective suits | 1.990 | .503 |
| face shields | 1.553 | .644 |

three groups of multivariate linear regression models were examined. The first model group examined the association of PPE and different PPE scores with PP. The second examined knowledge of a pandemic plan, and the third explored estimates of the utility of known pandemic plans on PP.

In the first set of linear regressions (Table 3), we examined PPE more closely. In the first linear regression model, the association between a general PPE score and PP was measured with age and gender as control variables. The PPE score showed to be a significant coefficient with a positive, non-standardized effect of 1.011 on PP; the adjusted $R^2$ value was .348. In the second model, a masks-only score was calculated on the basis of only the two mask types (FFP-2/3 and medical masks). The masks-only score explained the variance in the data analogously to the general PPE score in the first model (adj. $R^2$ = .349), with a significant association of .797 (p < .001). All PPE items were included individually in the third model (Table 3). Because 61 physicians did not provide information on all PPE items, only 445 responses from GPs were included into this model. The model with the individual materials showed an adjusted $R^2$ of .359. FFP-2/3 masks (coef. = .263), medical masks (coef. = .252), protective suits (coef. = .229), and face shields (coef. = .207) had a significant positive effect on PP. On the basis of these exploratory findings, an optimized PPE score was generated in model 4 (Table 3), with the four significant PPE items. This score was found to explain the observed variances slightly better (adj. $R^2$ = .379) than the general PPE score or the masks-only score, with a significant effect of .928 (p < .001). In the first two models (general PPE score and masks-only score), the control variable age >60 was significantly negatively associated with PP.

The next set of models (Table 4) examined the effect of knowledge of a pandemic plan on perceived PP. The first linear regression model (Table 4) included the variable regarding knowledge of a pandemic plan (no = 0, yes = 1) and the two control variables age and gender. A significant positive association of .521 with PP was found; however, the model quality was low, with an adjusted $R^2$ of .037. Adding the general PPE score containing all PPE items to the model (model 2, Table 4) improved the explained variance, with an adjusted $R^2$ value of .359. Both PPE score (coef. = .987) and knowledge of a pandemic plan (coef. = .300) are significantly positively associated with perceived PP. Similar results were observed when using the masks-only score (model 3, Table 4) as well as the optimized PPE score (model 4, Table 4). In addition, the covariance factor for age >60 showed significant associations with PP in the model with the masks-only score. The final variable included in the model was an interaction term between the general PPE score and knowledge on a pandemic plan (model 5, Table 4). The interaction term showed no association with perceived PP. Likewise, there was no significant interaction effect observed in the other two models with the other two PPE scores. These two models are not presented here.

In the final group of models (Table 5), we considered only those GPs who reported being aware of a pandemic plan prior to the outbreak of the COVID-19 pandemic. These 70 GPs

**Table 3. Multivariable linear regression model on personal protective equipment and pandemic preparedness among general practitioners.**

| Parameter | | Model I (General PPE score) | | | Model II (Masks score) | | | Model III (individual PPE materials) | | | Model IV (optimized PPE score) | | |
|---|---|---|---|---|---|---|---|---|---|---|---|---|---|
| | | Estimate [95% conf. interval] | Std. error | P-value | Estimate [95% conf. interval] | Std. error | P-value | Estimate [95% conf. interval] | Std. error | P-value | Estimate [95% conf. interval] | Std. error | P-value |
| Intercept | | .227 [-.109; .563] | .171 | .185 | .913 [.636; 1.190] | .141 | < .001 | .472 [.057; .887] | .211 | .026 | .785 [.509; 1.060] | .140 | < .001 |
| Independent variables | | | | | | | | | | | | | |
| | FFP-2/3 masks | | | | | | | **.263 [.136; .390]** | **.065** | **< .001** | | | |
| | medical face masks | | | | | | | **.252 [.147; .354]** | **.053** | **< .001** | | | |
| | disposable gloves | | | | | | | .058 [-.083; .200] | .072 | .419 | | | |
| | hand and surface disinfectants | | | | | | | .077 [-.051; .204] | .065 | .238 | | | |
| | safety glasses | | | | | | | -.048 [-.151; .055] | .052 | .356 | | | |
| | protective suits | | | | | | | **.229 [.096; .363]** | **.068** | **.001** | | | |
| | face shields | | | | | | | **.207 [.081; .333]** | **.064** | **.001** | | | |
| | PPE-Score | **1.011 [.889; 1.133]** | **.062** | **< .001** | | | | | | | | | |
| | PPE-Score optimized | | | | | | | | | | **.928 [.822; 1.033]** | **.053** | **< .001** |
| | Masks-Score | | | | **.797 [.701; .894]** | **.049** | **< .001** | | | | | | |
| Control variables | | | | | | | | | | | | | |
| | Age | | | | | | | | | | | | |
| | *41 to 50 years* | -.140 [-.388; .108] | .126 | .267 | -.128 [-.374; .118] | .125 | .308 | -.161 [-.407; .084] | .125 | .198 | -.116 [-.356; .124] | .122 | .343 |
| | *51 to 60 years* | -.141 [-.376; .094] | .120 | .238 | -.195 [-.428; .039] | .119 | .102 | -.207 [-.438; .023] | .117 | .078 | -.127 [-.355; .101] | .116 | .273 |
| | *older than 60 years* | -.185 [-.438; .067] | .128 | .150 | **-.258 [-.509; -.007]** | **.128** | **.044** | **-.262 [-.514; -.010]** | **.128** | **.041** | -.192 [-.437; .054] | .125 | .125 |
| | Gender | | | | | | | | | | | | |
| | *female* | -.010 [-.152; .132] | .072 | .887 | -.066 [-.206; .075] | .072 | .360 | -.046 [-.191; .099] | .074 | .537 | -.066 [-.203; .072] | .070 | .348 |
| Number of obs. | | 506 | | | 505 | | | 445 | | | 505 | | |
| $R^2$ | | .355 | | | .355 | | | .375 | | | .385 | | |
| Adj. $R^2$ | | .348 | | | .349 | | | .359 | | | .379 | | |
| F-stats | | 54.986 | | | 54.994 | | | 23.623 | | | 62.568 | | |
| df | | 500 | | | 499 | | | 433 | | | 499 | | |
| p-value | | < .001 | | | < .001 | | | < .001 | | | < .001 | | |

were asked whether they considered the known pandemic plan beneficial in managing the COVID-19 pandemic. The first regression model (model 1, Table 5) showed that the assessment of the pandemic plan as beneficial controlled for the two variables age and gender was not significantly associated with perceived PP. In the next model (model 2, Table 5), the PPE score was added. With the addition of the PPE score, the explanation of variance increased to an adjusted $R^2$ of .546; again, the PPE score itself showed a significant positive association with

**Table 4. Multivariable linear regression model on knowledge of a pandemic plan and pandemic preparedness among general practitioners.**

| Parameter | | Model I (Knowledge on plan) | | | Model II (Knowledge on plan + PPE score) | | | Model III (Knowledge on plan + masks score) | | | Modell IV (Knowledge on plan + optimized PPE score) | | | Model V (Knowledge on plan:PPE score) | | |
|---|---|---|---|---|---|---|---|---|---|---|---|---|---|---|---|---|
| | | Estimate [95% conf. interval] | Std. error | P-value | Estimate [95% conf. interval] | Std. error | P-value | Estimate [95% conf. interval] | Std. error | P-value | Estimate [95% conf. interval] | Std. error | P-value | Estimate [95% conf. interval] | Std. error | P-value |
| Intercept | | 2.309 [2.046; 2.571] | .134 | < .001 | .240 [-.094; .574] | .170 | .158 | .895 [.620; 1.170] | .140 | < .001 | .780 [.507; 1.054] | .139 | < .001 | .341 [.014; .696] | .180 | .059 |
| Independent variables | | | | | | | | | | | | | | | | |
| | Prior knowledge of pandemic plan | **.521 [.278; .764]** | **.124** | **< .001** | **.300 [.100; .500]** | **.102** | **.003** | **.333 [.135; .532]** | **.101** | **.001** | **.274 [.079; .469]** | **.099** | **.006** | -.265 [-.971; .442] | .360 | .462 |
| | PPE score | | | | **.987 [.865; 1.109]** | **.062** | **< .001** | | | | | | | **.936 [.800; 1.073]** | **.069** | **< .001** |
| | PPE score optimized | | | | | | | | | | **.909 [.804; 1.015]** | **.054** | **< .001** | | | |
| | Mask score | | | | | | | **.783 [.687; .879]** | **.049** | **< .001** | | | | | | |
| | Knowledge: PPE score | | | | | | | | | | | | | .253 [-.050; .555] | .154 | .102 |
| Control variables | | | | | | | | | | | | | | | | |
| | Age | | | | | | | | | | | | | | | |
| | *41 to 50 years* | -.050 [-.351; .252] | .153 | .746 | -.147 [-.393; .099] | .125 | .242 | -.136 [-.380; .108] | .124 | .273 | -.123 [-.361; .116] | .121 | .313 | -.139 [-.385; .107] | .125 | .266 |
| | *51 to 60 years* | -.150 [-.436; .136] | .146 | .304 | -.141 [-.375; .093] | .119 | .236 | -.191 [-.423; .040] | .118 | .105 | -.128 [-.355; .099] | .115 | .268 | -.140 [-.373; .093] | .119 | .239 |
| | *older than 60 years* | -.304 [-.611; .003] | .156 | .052 | -.200 [-.450; .051] | .128 | .119 | **-270 [-.518; -.021]** | **.127** | **.034** | -.203 [-.446; .041] | .124 | .103 | -.194 [-.445; .056] | .127 | .129 |
| | Gender | | | | | | | | | | | | | | | |
| | *female* | -.088 [-.260; .085] | .088 | .319 | -.010 [-.151; .131] | .072 | .893 | -.060 [-.200; .079] | .071 | .397 | -.063 [-.200; .073] | .070 | .363 | -.010 [-.151; .131] | .072 | .891 |
| Number of obs. | | 504 | | | 504 | | | 503 | | | 503 | | | 504 | | |
| R² | | .046 | | | .367 | | | .369 | | | .395 | | | .370 | | |
| Adj. R² | | .037 | | | .359 | | | .362 | | | .388 | | | .361 | | |
| F-stats | | 4.843 | | | 47.968 | | | 48.409 | | | 54.085 | | | 41.638 | | |
| df | | 498 | | | 497 | | | 496 | | | 496 | | | 496 | | |
| p-value | | < .001 | | | < .001 | | | < .001 | | | < .001 | | | < .001 | | |

PP (1.195), but an assessment of the utility of the pandemic plan did not. In the final regression model (model 3, Table 5), an interaction term was formed between the assessment of the pandemic plan as beneficial and the PPE score. Assessing the pandemic plan as beneficial did not

**Table 5. Multivariable linear regression model of assessment of pandemic plan as beneficial and pandemic preparedness among general practitioners.**

| Parameter | | Model I (Plan helpful) | | | Model II (Plan helpful + PPE-Score) | | | Model III (Plan helpful + PPE-Score + Plan helpful:PPE-Score) | | |
|---|---|---|---|---|---|---|---|---|---|---|
| | | Estimate [95% conf. interval] | Std. error | P-value | Estimate [95% conf. interval] | Std. error | P-value | Estimate [95% conf. interval] | Std. error | P-value |
| Intercept | | 2.717 [1.830; 3.603] | .444 | < .001 | -.207 [-1.079; .665] | .436 | .637 | -.668 [-1.639; .304] | .486 | .174 |
| Independent variables | | | | | | | | | | |
| | Pandemic plan helpful | .348 [-.196; .892] | .272 | .206 | .300 [-.062; .661] | .181 | . 102 | **1.476 [.236; 2.715]** | **.620** | **.020** |
| | PPE score | | | | **1.195 [.932; .459]** | **.132** | < **.001** | **1.404 [1.071; 1.737]** | **.166** | < **.001** |
| | Pandemic plan helpful: PPE-Score | | | | | | | -.515 [-1.035; .005] | .260 | .052 |
| Control variables | | | | | | | | | | |
| | Age | | | | | | | | | |
| | *41 to 50 years* | -.090 [-1.067; .887] | .489 | .854 | .038 [-.611; .687] | .325 | .907 | .023 [-.611; .658] | .318 | .941 |
| | *51 to 60 years* | -.032 [-.991; .927] | .480 | .947 | .030 [-.606; .666] | .319 | .925 | .026 [-.596; .649] | .311 | .933 |
| | *older than 60 years* | -.262 [-1.245; .721] | .492 | .597 | .100 [-.557; .758] | .329 | .761 | .118 [-.525; .761] | .322 | .716 |
| | Gender | | | | | | | | | |
| | *female* | -.237 [-.792; .318] | .278 | .396 | -.100 [-.470; .269] | .185 | .590 | -.124 [-.486; .238] | .181 | .495 |
| Number of obs. | *504* | 70 | | | 70 | | | 70 | | |
| $R^2$ | | *.042* | | | *.585* | | | *.610* | | |
| Adj. $R^2$ | | *-.032* | | | *.546* | | | *.566* | | |
| F-stats | | *.566* | | | *14.805* | | | *13.837* | | |
| df | | *64* | | | *63* | | | *62* | | |
| p-value | | *.725* | | | < .001 | | | < .001 | | |

significantly interact with the PPE score on perceived PP (p = .052). The model indicated a good fit, with an adjusted $R^2$ of .566. Models with the other two scores did not show such close significance values. These two models are not presented here.

## Discussion

The aim of this study was to determine the association of stockpiled PPE and knowledge of pandemic plans on the PP of German GPs. It has been shown that the stock of PPE is the most important factor for PP. Different PPE scores differed only to a small extent in the variance explained. Knowledge of a pandemic plan also showed to be significantly associated with PP, but the association was much smaller in comparison with PPE. Assessment of the utility of a known pandemic plan showed no significant association with PP.

Numerous studies on the effect of the COVID-19 pandemic on the outpatient sector report low levels of PP in Germany [37] and in several other countries [8,38], with only a few exceptions [10]. The significance of availability and access to PPE for pandemic management was frequently observed during the COVID-19 pandemic [7,8] and during other pandemics [39]. However, many studies do not specifically address the particular inventory of different PPE items [37]. Our results suggest that for reasons of simplicity and data minimization, it seems appropriate to focus on the stock of FFP-2/3 and medical masks in regard to PP in the context of the COVID-19 pandemic. The insignificant changes in the explained variance of the different PPE scores point in this direction. Because SARS-COV-2 is transmitted via the respiratory tract, this focus seems theoretical plausible as well. However, comparability between the

different PPE scores and the model with individual PPE items is somewhat limited by missing individual values for different items. In particular, face shields were not considered relevant by 57 GPs. This high number of assessments of face shields as irrelevant contradicts to some extent the results of our model, where a significant positive association between face shields and PP was identified. Also other studies have shown that eye and face protection are important factors [40]. The significant positive association of protective suits cannot be classified in the category of protection of eyes and face. Because the survey referred to the beginning of the pandemic in Germany in March–April 2020, this may can be interpreted as the effect of a great uncertainty among the GPs, who demanded complete protection on the face of great uncertainty.

The calculation of the mean value of the PPE scores was chosen in order to consider possible interactions between different PPE items. However, the results between the models with PPE scores and the model with the individual PPE items did not show large differences for the different approaches. Thus, an actual interaction between different PPE items has not been confirmed beyond doubt. It can also be argued that a simple average of PPE items does not adequately represent the interaction. It may would be conceivable to weight lower inventories to a greater extent. Different PPE items are needed for optimal protection, so the lack of just one item may make sufficient stocks in all other items inadequate.

Physicians face unique challenges in times of pandemics; therefore, a well-structured and widely known pandemic plan is believed to help establish effective strategies in advance [8]. However, we found that the majority of GPs considered a pandemic plan not beneficial regarding the COVID-19 pandemic and that the assessment of a plan as beneficial did not show a significant effect on PP, whereas pre-existing knowledge of such a plan had a small positive effect on PP, which indicates that the specific content of the pandemic plan is somewhat less relevant. If engagement with a pandemic plan helps to address general and cross-pandemic processes in advance, it may create an overall pandemic awareness that can be adapted to individual challenges of a particular pandemic. Knowledge on a pandemic plan could than serve as proxy for pandemic awareness. Nevertheless, the variance explained by this predictor was rather small. However, the knowledge of a pandemic plan may also have an opposite effect of decreasing the perceived PP because the knowledge of such plans makes GPs aware of what they have to consider and how great their deficits truly are. Furthermore, other influencing factors of PP not examined here, include profound knowledge about the disease and the manner in which that knowledge is disseminated [41] and proper use of PPE [39], fear of transmitting the infection to families and loved ones [39], compliance of healthcare workers with proper infection prevention [42], emotional support [43], and years of experience, and training in infection control [44].

The interaction term examining the relationship between the assessment of the pandemic plan as beneficial and the stockpile of PPE items in model 3 in Table 4 showed a non-significant association between the assessment of a pandemic plan as beneficial and the PPE score that was just slightly above the threshold for significance at .05 (p = .052). Because only 70 GPs were even aware of a pandemic plan, this association should be further investigated. Our findings give rise to the hypothesis that when a pandemic plan is considered beneficial, the quantity of PPE items is not quite as crucial as without this assessment. Knowledge of a beneficial pandemic plan would than enhance the effect on perceived PP when the PPE items is in low supply, but when sufficient PPE are available, the positive effect of PPE on PP is no longer quite as large.

## Limitations

As the cross-sectional online survey was conducted in the early stages of the COVID-19 pandemic in Germany, the study may have certain limitations. First, the survey was conducted in

June–September 2020 retrospectively for the period March–April 2020. Therefore, the possibility that evaluations and assessments were ex post distorted between the observation and survey period cannot be eliminated, especially in the case of a dynamic event such as a pandemic. The survey period was chosen in order to consider the different summer school holidays in the German federal states. Second, although the sample was chosen for representative purposes, selection bias may have occurred owing to the low response rate and the distribution of the survey via the project homepage and the different specialist societies, which makes it challenging to draw conclusions about all German GPs. The low response rate may be explained by GPs' increased workload and uncertainty during a pandemic. Because PP among German GPs was generally rated as poor and this is a cross-sectional study, it is not clear whether there is a true causal relationship between stockpiled PPE and perceived PP. Moreover, the results on individual PPE items may have limited applicability to other pandemic scenarios as each pandemic presents different challenges to physicians and the infection and transmission pathways differ between pandemics. With regard to the provision of disinfectants, the inventory of hand and surface disinfectants was queried together. Accordingly, this survey cannot provide more precise information on the distinction between the two PPE materials.

Because this is an anonymous survey, it cannot be ruled out that participants may have responded to the survey more than once or that non-physicians participated. Though, at the beginning of the survey, it was asked whether the participant works as a physician in the outpatient sector. If this answer was negative, participation in the survey was terminated. Nevertheless, it cannot be ruled out that deliberately false statements were made here. The selected recruitment method does not allow representative conclusions for GPs in Germany. However, if we consider key sociodemographic characteristics of the study participants (S2 Table) and compare them with the basic population of German GPs, it becomes clear that there are indications that the study population represents German GPs reasonably accurately. First of all, participants from all 16 German federal states and city states took part in the study. In view of the statistical data from the German Federal Register of Physicians, it appears that our study population was, on average, somewhat younger than the average German GPs (approximately 53 years compared to 55.4 years) [45]. With regard to the gender distribution of the sample, this corresponds to the national average for GPs (52% male, 48% female) provided by the Federal Register of Physicians [46]. About 90% of the physicians surveyed reported that they are self-employed. This is about 10 percentage points higher than the national average according to data from the 2020 physician statistics of the German Medical Association [47]. The overrepresentation of self-employed physicians can possibly be explained by the fact that they were contacted via fax. Although the invitation letters were personalized, the faxes may nevertheless have been presented to the practice owner. Also, in the case of practice email addresses, the practice owner may have been the primary contact or may have had access first. It is also possible that self-employed physicians have a higher level of commitment and identification with their own profession, so that a slight selection bias cannot be ruled out. Furthermore, a possible selections bias may also have implications for the reported PP. More job committed individuals may also have higher general preparedness. As a result, this could lead to a slight overestimation of the pandemic preparedness of the analyses in the population.

## Conclusion

In Germany, a large proportion of GPs believed that they were poorly or very poorly prepared for a pandemic at the beginning of the COVID-19 pandemic; however, high PP among GPs can play a vital role in ensuring that the healthcare sector as a whole is better prepared for future pandemics. Pandemic preparedness can be explained in large part by the possession of

sufficient PPE. Possession of FFP-2/3 masks, medical masks, protective suits, and face shields are significantly positively associated with PP. The findings of the study justify focusing on the stock of medical and FFP 2/3 masks among PPE. Overall, only 14% of GPs had knowledge about a pandemic plan. A multivariate linear regression analysis showed that knowledge of a pandemic plan is significantly associated to a small positive extent with perceived PP among German GPs. However, the positive association of PPE significantly exceeded that of knowledge of a pandemic plan. Whether the known pandemic plan was rated as beneficial or not showed no effect on addressing the challenges associated with COVID-19. The PP of German GPs thus depends largely on the stockpile of PPE; pandemic plans play a rather subordinate role.

## Supporting information

**S1 Table. Questionnaire (German and English translation) questionnaire (German, original version).**
(DOCX)

**S2 Table. Key sociodemographic characteristics.**
(DOCX)

## Acknowledgments

The authors express their gratitude to all participating physicians. Despite the fact that they remain anonymous, this study would not have been possible without their participation.

## Author Contributions

**Conceptualization:** Arno Stöcker, Nadine Scholten.

**Data curation:** Sophie Gunkel.

**Formal analysis:** Arno Stöcker.

**Funding acquisition:** Arno Stöcker, Tim Ohnhäuser, Nadine Scholten.

**Investigation:** Arno Stöcker, Sophie Gunkel, Jan Hoffmann, Laura Mause, Tim Ohnhäuser, Nadine Scholten.

**Methodology:** Arno Stöcker, Ibrahim Demirer, Nadine Scholten.

**Project administration:** Arno Stöcker, Tim Ohnhäuser.

**Supervision:** Nadine Scholten.

**Writing – original draft:** Arno Stöcker.

**Writing – review & editing:** Ibrahim Demirer, Jan Hoffmann, Laura Mause, Nadine Scholten.

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
