## [Decision Letter · Decision Letter 0]

11 Jun 2021

PONE-D-21-13972

Stockpiled personal protective equipment and knowledge of pandemic plans as predictors of perceived pandemic preparedness among German general practitioners

PLOS ONE

Dear Dr. Stöcker,

Thank you for submitting your manuscript to PLOS ONE. After careful consideration, we feel that it has merit but does not fully meet PLOS ONE’s publication criteria as it currently stands. Therefore, we invite you to submit a revised version of the manuscript that addresses the points raised during the review process.

We look forward to receiving your revised manuscript.

Kind regards,

Jianguo Wang, PhD

Academic Editor

PLOS ONE

Journal Requirements:

Furthermore, in your Methods section, please provide a justification for the sample size used in your study, including any relevant power calculations (if applicable).

3. Please provide additional details regarding participant consent. In the ethics statement in the Methods and online submission information, please ensure that you have specified (1) whether consent was suitably informed and (2) what type you obtained (for instance, written or verbal). If your study included minors under age 18, state whether you obtained consent from parents or guardians. If the need for consent was waived by the ethics committee, please include this information.

5. Please include a copy of Table 5 which you refer to in your text on page 15.

Reviewers' comments:

Reviewer's Responses to Questions

**Comments to the Author**

1. Is the manuscript technically sound, and do the data support the conclusions?

Reviewer #1: Yes

Reviewer #2: Yes

2. Has the statistical analysis been performed appropriately and rigorously? 

Reviewer #1: Yes

Reviewer #2: Yes

3. Have the authors made all data underlying the findings in their manuscript fully available?

Reviewer #1: Yes

Reviewer #2: Yes

4. Is the manuscript presented in an intelligible fashion and written in standard English?

Reviewer #1: Yes

Reviewer #2: Yes

5. Review Comments to the Author

Reviewer #1: Thank you for the chance to review this manuscript. I thought that this was generally a well-written manuscript focusing on the association between sufficiency of various PPE stocks and perceived pandemic preparedness among German GPs.

The authors found that PPE stocks in general seem to be significant associated with PP scores, and that facial protection in particular was a prominent piece of PPE. Both of these findings seem logical to me given the understanding of the COVID-19 situation and known transmission mechanics at the time that the study was conducted.

I also felt that the authors used appropriate regression analyses that were well detailed in the Results section, and the Discussion was able to present an interpretation without overclaiming the validity of the findings.

I just have two minor comments for the authors to improve their manuscript for potential publication:

1. I understand from your Methods section that 6,500 GPs were invited to participate, of which 535 GPs responded. This is a very low response rate of approximately 8.2%.

How might this low response rate have affected the generalisability of your study's findings? Given that your survey was online and anonymous, would it have been possible for certain selection biases (e.g., location, socio-economic status) to affect the findings? Is there a possibility of data errors (e.g., same GP responding to the survey more than once)?

If there is insufficient data (due to anonymity of the respondents) to establish the presence of these biases, it would be good to elaborate on them in the limitations section.

2. On line 121, you mention that the study population of GPs was relatively homogeneous as a way of justifying why only age and gender were chosen as potential confounders. It would be ideal for you to demonstrate to the reader that this homogeneity in your study sample is actually true. For example, you could present key sociodemographic characteristics of the sample in a descriptive table.

Reviewer #2: Hand hygiene is a very important part of Personal Protective Equipment - however in the manuscript data on hand hygiene presence (I assume alcohol-based handrub)is presented combined with surface disinfectant - the two should be distinguished from each other since different use and process and this change should be reflected in the result section also

In results Table 2 - can you clarify if the sanitizer is hand sanitizer?

Can you identify any psychometric testing/ validation/piloting performed of the survey instruments? What about validation of the survey instrument answers? Was there a sample of completed forms that were validated?

6. PLOS authors have the option to publish the peer review history of their article (what does this mean?). If published, this will include your full peer review and any attached files.

Reviewer #1: No

Reviewer #2: No

---

## [Author Response · Author response to Decision Letter 0]

23 Jun 2021

Dear reviewers,

Thank you for your valuable time and helpful comments on our manuscript. We have included a document entitled, "Response to Reviewers" with detailed information on how we addressed each of the comments.

Sincerely,

Arno Stöcker

---

## [Decision Letter · Decision Letter 1]

7 Jul 2021

PONE-D-21-13972R1

Stockpiled personal protective equipment and knowledge of pandemic plans as predictors of perceived pandemic preparedness among German general practitioners

PLOS ONE

Dear Dr. Stöcker,

Thank you for submitting your manuscript to PLOS ONE. After careful consideration, we feel that it has merit but does not fully meet PLOS ONE’s publication criteria as it currently stands. Therefore, we invite you to submit a revised version of the manuscript that addresses the points raised during the review process.

A minor revision is still required.

We look forward to receiving your revised manuscript.

Kind regards,

Jianguo Wang, PhD

Academic Editor

PLOS ONE

Journal Requirements:

Reviewers' comments:

Reviewer's Responses to Questions

**Comments to the Author**

1. If the authors have adequately addressed your comments raised in a previous round of review and you feel that this manuscript is now acceptable for publication, you may indicate that here to bypass the “Comments to the Author” section, enter your conflict of interest statement in the “Confidential to Editor” section, and submit your "Accept" recommendation.

Reviewer #1: All comments have been addressed

Reviewer #2: All comments have been addressed

2. Is the manuscript technically sound, and do the data support the conclusions?

Reviewer #1: Yes

Reviewer #2: Yes

3. Has the statistical analysis been performed appropriately and rigorously? 

Reviewer #1: Yes

Reviewer #2: Yes

4. Have the authors made all data underlying the findings in their manuscript fully available?

Reviewer #1: Yes

Reviewer #2: Yes

5. Is the manuscript presented in an intelligible fashion and written in standard English?

Reviewer #1: Yes

Reviewer #2: Yes

6. Review Comments to the Author

Reviewer #1: Thank you for adequately addressing my suggested amendments to your manuscript.

I have just one more minor comment:

Limitations (page 21)

- Thank you for elaborating in detail about the limitations of an anonymous online survey, as well as the attempt to relate your sample's sociodemographic distribution to national demographic sources for German GPs. However, I would suggest that you cite the respective sources (e.g., Federal Register of Physicians, German Medical Association). Otherwise it is difficult for the reader to access and compare the relevant demographic characteristics accordingly.

Reviewer #2: (No Response)

7. PLOS authors have the option to publish the peer review history of their article (what does this mean?). If published, this will include your full peer review and any attached files.

Reviewer #1: No

Reviewer #2: No

---

## [Author Response · Author response to Decision Letter 1]

24 Jul 2021

Dear Reviewers,

Thank you again for your valuable comment. Please find enclosed a revised version of our original research article for publication in PLoS ONE. We have included a document entitled, "Response to Reviewers" with detailed information on how we addressed your remaining comment.

Thank you for your time and effort.

Sincerely,

Arno Stöcker

---

## [Decision Letter · Decision Letter 2]

28 Jul 2021

Stockpiled personal protective equipment and knowledge of pandemic plans as predictors of perceived pandemic preparedness among German general practitioners

PONE-D-21-13972R2

Dear Dr. Stöcker,

We’re pleased to inform you that your manuscript has been judged scientifically suitable for publication and will be formally accepted for publication once it meets all outstanding technical requirements.

Kind regards,

Jianguo Wang, PhD

Academic Editor

PLOS ONE

Additional Editor Comments (optional):

Reviewers' comments:

Reviewer's Responses to Questions

**Comments to the Author**

1. If the authors have adequately addressed your comments raised in a previous round of review and you feel that this manuscript is now acceptable for publication, you may indicate that here to bypass the “Comments to the Author” section, enter your conflict of interest statement in the “Confidential to Editor” section, and submit your "Accept" recommendation.

Reviewer #1: All comments have been addressed

2. Is the manuscript technically sound, and do the data support the conclusions?

Reviewer #1: Yes

3. Has the statistical analysis been performed appropriately and rigorously? 

Reviewer #1: (No Response)

4. Have the authors made all data underlying the findings in their manuscript fully available?

Reviewer #1: Yes

5. Is the manuscript presented in an intelligible fashion and written in standard English?

Reviewer #1: Yes

6. Review Comments to the Author

Reviewer #1: Thank you for addressing my concerns. I have no further comments, and wish the authors all the best!

7. PLOS authors have the option to publish the peer review history of their article (what does this mean?). If published, this will include your full peer review and any attached files.

Reviewer #1: No

---

## [Editor Report · Acceptance letter]

3 Aug 2021

PONE-D-21-13972R2 

Stockpiled personal protective equipment and knowledge of pandemic plans as predictors of perceived pandemic preparedness among German general practitioners 

Dear Dr. Stöcker:

I'm pleased to inform you that your manuscript has been deemed suitable for publication in PLOS ONE. Congratulations! Your manuscript is now with our production department. 

Kind regards, 

on behalf of

Dr. Jianguo Wang 

Academic Editor

PLOS ONE